# The Role of the Large T Antigen in the Molecular Pathogenesis of Merkel Cell Carcinoma

**DOI:** 10.3390/genes15091127

**Published:** 2024-08-27

**Authors:** Julia Myrda, Franziska Bremm, Niels Schaft, Jan Dörrie

**Affiliations:** 1Department of Dermatology, Universitätsklinikum Erlangen, Friedrich-Alexander-Universität Erlangen-Nürnberg, 91054 Erlangen, Germany; julia.myrda@extern.uk-erlangen.de (J.M.); franziska.bremm@uk-erlangen.de (F.B.); niels.schaft@uk-erlangen.de (N.S.); 2Comprehensive Cancer Center Erlangen European Metropolitan Area of Nuremberg (CCC ER-EMN), 91054 Erlangen, Germany; 3Deutsches Zentrum Immuntherapie (DZI), 91054 Erlangen, Germany; 4Bavarian Cancer Research Center (BZKF), 91054 Erlangen, Germany

**Keywords:** Merkel cell carcinoma, Merkel cell polyomavirus, viral carcinogenesis, antigen-based immunotherapy, cell cycle regulation, immune escape

## Abstract

The large T antigen (LT) of the Merkel cell polyomavirus (MCPyV) is crucial for Merkel cell carcinoma (MCC), a rare but very aggressive form of neuroendocrine skin cancer. The clonal integration of MCPyV DNA into the host genome is a signature event of this malignancy. The resulting expression of oncogenes, including the small T (sT) antigen and a truncated form of the LT (truncLT), directly contribute to carcinogenesis. The truncation of the C-terminus of LT prevents the virus from replicating due to the loss of the origin binding domain (OBD) and the helicase domain. This precludes cytopathic effects that would lead to DNA damage and ultimately cell death. At the same time, the LxCxE motif in the N-terminus is retained, allowing truncLT to bind the retinoblastoma protein (pRb), a cellular tumor suppressor. The continuously inactivated pRb promotes cell proliferation and tumor development. truncLT exerts several classical functions of an oncogene: altering the host cell cycle, suppressing innate immune responses to viral DNA, causing immune escape, and shifting metabolism in favor of cancer cells. Given its central role in MCC, the LT is a major target for therapeutic interventions with novel approaches, such as immune checkpoint inhibition, T cell-based immunotherapy, and cancer vaccines.

## 1. Introduction to Merkel Cell Carcinoma

Merkel cell carcinoma (MCC) is a rare and highly aggressive form of skin cancer with a 5-year overall survival rate of around 40% and the highest case-by-case fatality rate among all skin cancers, and it predominantly affects elderly and immunosuppressed patients [1,2,3,4,5]. Because of its typical clinical phenotype, Cyril Toker described the tumor in 1972 as ‘trabecular carcinoma of the skin’ and characterized it by the rapid growth of malignant cells in the neuroendocrine cells of the dermis [6]. It usually presents as cutaneous or subcutaneous pink-red dome-shaped nodules, most commonly in the head and neck region, extremities, or in the periocular region, with a high prevalence of lymphatic metastasis [1,7,8].

The neurosecretory cytoplasmic granules in Tokers ‘trabecular carcinoma of the skin’ [6] resemble those found in Merkel cells, and therefore, Merkel cell carcinoma was named after its originally presumed cell of origin [9,10]. However, in recent years, studies of the genetic and morphological characteristics of MCC have questioned Merkel cells as the cell of origin of MCC [11] and suggested, among others, epithelial stem cells, pre-/pro-B cells, dermal mesenchymal cells, or cells of the neuronal lineage, offering different hypotheses for tumorigenesis [1,2].

While ultraviolet (UV) radiation has been known for decades to be the primary cause of skin cancers such as malignant melanoma [12], squamous cell carcinoma [13,14,15], or basal cell carcinoma [12,13,14,15], the observation that MCC occurs disproportionately often in elderly and immunocompromised patients led to the suggestion that the causative agent might be of viral origin [1]. Through the transcriptomic analysis of MCC tissue, the Merkel cell polyomavirus (MCPyV), a DNA virus from the family of polyomaviruses, was discovered [1,16,17]. In approximately 80% of MCC cases in the USA and Europe, MCPyV can be traced as the disease-causing agent, whereas the majority of MCC in Australia is MCPyV-negative and is associated with chronic UV-light exposure [8,17,18,19,20,21]. Despite this dichotomy, both forms appear clinically and phenotypically very similar and share histological and prognostic features [1,8,11]. While MCPyV-positive MCC is characterized by a low mutation burden, MCPyV-negative MCC is associated with the presence of UV-light-induced DNA damage (C > T mutations at dipyrimidine sites), a high number of copy number variations, and a high burden of somatic modifications (>20 mutations/Mb) [22,23,24]. Recurrent mutations particularly occur in tumor suppressor genes including *TP53* (7/8, 87.5%) and *RB1* (5/8, 62.5%) [23].

Although there is a high seroprevalence of MCPyV in adults, mostly with asymptomatic infections [25,26,27,28], only in a few people does the viral oncogene, the large T antigen (LT), induce tumorigenesis. The LT needs to acquire a specific mutational truncation and integrate into the human genome to cause cancer. The expression of the truncated form of LT (truncLT) induces uncontrolled cell proliferation, while the truncation prevents viral replication. Due to the stable integration, all daughter cells express truncLT, which causes tumorigenesis in this tissue. Understanding the molecular processes that induce this oncogenic transformation of MCPyV-infected cells can help to provide new targets for the treatment of MCC patients. Therefore, in this review, we want to summarize the knowledge of the MCPyV itself, the role of the viral oncogene LT in tumorigenesis, and the immune escape mechanisms of virus-positive MCC, as well as give a short overview of treatment strategies that are based on the LT antigen specifically.

## 2. The Merkel Cell Polyomavirus

### 2.1. General Properties of the Virus

The MCPyV that causes MCC is a polyomavirus similar to the simian vacuolating virus 40 (SV40) [29]. MCPyV is a small non-enveloped, double-stranded DNA (dsDNA) virus with a genome of approximately 5400 base pairs [29,30,31]. Studies suggest a seroprevalence of up to 80% in adults and up to 45% in children [25,26,27,30]. Considered as part of a healthy human skin virome [27,30], the virus persist in healthy individuals episomally during the infection cycle, and the infection remains asymptomatic in most cases [28].

The genome of MCPyV can be divided into the opposing early regions (ER) and late regions (LR) (Figure 1A) [30,31]. The region in between is termed the noncoding regulatory region (NCRR) and contains the origin of replication (ori) where bi-directional promoters are required for viral transcription [31]. Dividing the ER from the LR provides a temporal containment between early and late gene expression that can be characterized by the onset of DNA replication [29]. The gene products of the early region are directly involved in viral DNA replication and are therefore expressed immediately after infection. With about three kilobases, the ER genes account for the largest proportion of the viral genome and contain all the tumor antigens, also referred to as the Tumor antigen or T antigen (TA) locus [29]. Immediately after infection, the TA is transcribed, which encodes distinct non-structural T proteins, the LT, the small TA (sT), and the 57 kT antigen (Figure 1B) [32,33,34]. Although the transcript is spliced alternatively, all three antigens share the first exon sequence [32,33,34]. In addition, the TA locus encodes a protein from an alternate reading frame called ALTO [33,34].

MCPyV infects the host cell mainly via the viral capsid protein (VP1) by binding to sulfated and sialylated glycoconjugates, which act as entry factors [1,35]. Similar to other known polyomaviruses, viral internalization is then mediated by caveolar/lipid raft-dependent endocytosis with subsequent transport to endosomes that end in the endoplasmic reticulum. The endoplasmic reticulum then initiates capsid uncoating before the viral genome is transported via nuclear pore complexes into the nucleus, where the ER genes of MCPyV, especially the TAs, are expressed [1]. Subsequently, LT accumulates in the cell. This triggers the replication of the viral genome, which in turn induces the expression of the genes of the LR, which encode the capsid proteins (VP1 and VP2) required to assemble new virions (Figure 2) [30,36].

### 2.2. The Large T Antigen

The LT gene of MCPyV is similar to the LT of other polyomaviruses and contains multiple functional domains (Figure 1C): a DnaJ domain, two cysteine (CxCxxC) motifs, a retinoblastoma (Rb) binding motif called LxCxE, a threonine-proline-proline-lysine (TPPK) motif, a nuclear localization signal (NLS), and an origin binding domain (OBD). At the C-terminus, there is additionally a helicase [5,37].

The amino-terminal DnaJ domain is known to be critical for replication, although the molecular mechanism behind this function is not fully understood [30,37]. Adjacent to the DnaJ domain are two CxCxxC motifs, which are only expressed in the sT transcript and bind protein phosphatase 2A (PP2A) [38,39].

Many polyomaviruses bind benzimidazole 1 (BUB1), insulin receptor substrate 1 (IRS1), and cullin 7 (CUL7) in the domain between DnaJ and LxCxE (Figure 1C) [5,37,40]. Here, the LT of MCPyV contains a unique domain compared to other polyomaviruses, which was therefore named Merkel cell polyomavirus T-antigen unique region (MUR) by Liu et al. [40]. BUB1 is a checkpoint monitoring assembly of the mitotic spindle, but in MCPyV, the binding motif for BUB1 is only partially present, and BUB1 seems to have no interaction with the LT [40,41,42]. IRS1 plays a role in the transformation of cells by SV40 [43], but such a connection has yet to be demonstrated for MCPyV. CUL7 is associated with the targeted ubiquitination of proteins leading to a proteasomal degradation [44,45]. Furthermore, as an E3 ligase, CUL7 is able to bind p53, which reduces the function of this tumor suppressor protein [46]. Although the T antigen of SV40 interacts with CUL7 [47,48], such a relation has not yet been shown for MCPyV.

While other polyomaviruses bind CUL7, BUB1, and IRS1, the MUR of MCPyV T antigen binds to human Vam6p, which is homologous to the vacuolar protein sorting 39 protein (Vps39) in *Saccharomyces cerevisiae* [40]. The protein hVam6p is usually involved in lysosomal trafficking in the cytoplasm, but upon binding to LT, hVam6p translocates into the nucleus, where it loses its function [40].

Next to the MUR is the Rb-binding LxCxE motif, which is ubiquitous in all human polyomaviruses and is required for the disruption of Rb protein (pRb) functions by direct binding to pRb [37]. The pRb-like proteins p130 (RBL2) and p107 (RBL1) also bind to this region of the LT [37]. The pRb proteins have a tumor-suppressing function via their interference with the transactivating activity of E2F transcription factors. In cooperation with the DnaJ domain, binding to the LxCxE motif of the LT prevents pRb from inhibiting the expression of E2F target genes [24,37,49,50]. Furthermore, LT recruits the heat shock cognate 71 kDa protein (HSP70), a chaperone that is highly expressed in cancer cells and is known to be required for the growth of both cancer cells and normal cells [51]. HSP70 is activated upon cellular stress and is involved in folding and unfolding proteins [36]. By binding to the non-phosphorylated form of the pRb, HSP70 stabilizes pRb, but it can also interact with RBL2 and RBL1 [36,51]. In MCPyV-infected cells, LT recruits HSP70, which disrupts the complex formation between pRb and E2F. The released E2F then promotes entry into the S phase of the cell cycle, driving cell cycle progression and cell proliferation [5,24,36].

Adjacent to the LxCxE motif is a TPPK motif, which is important for viral replication upon phosphorylation of the threonine residue [37]. The NLS adjacent to TPPK mediates nucleocytoplasmic trafficking by binding to karyopherin-α (KPNA) family importin homologues [37]. The C-terminus of LT contains the OBD, which is a DNA binding domain that recognizes the LT binding site in the ori [52,53,54]. The helicase domain in the C-terminus with its ATP binding domain is critical for viral DNA replication and works together with the OBD [36,37,53,55,56]. The helicase domain of the TA gene of many polyomaviruses contains a direct p53 binding sequence. p53 is a tumor suppressor gene that can activate DNA repair proteins, arrest the cell cycle, and induce senescence or apoptosis to prevent abnormal cell growth when exposed to stressors [5,30,57]. It has been shown that the full-length LT of MCPyV with functional helicase activity caused the cell cycle to arrest, leading to limited cell proliferation in a means dependent on p53 [30]. But in contrast to SV40, BK polyomavirus (BKPyV) and JC polyomavirus (JCPyV) inhibit p53 by directly binding to their LT, and the full-length LT of MCPyV does not interact directly with the tumor suppressor [58,59]. Instead, MCPyV ensures the prolongation of its infection cycle by avoiding the activation of p53 through a low expression of the LT.

The replication of the viral genome is similar in all known human polyomaviruses and depends on the LT [30]. The helicase domain of LT contains a conserved ATP-binding site that facilitates helicase-mediated hydrolysis, which is crucial to inducing a conformational change in the LT protein [36,55]. This allows a bilobed double hexameric helicase complex of twelve LT monomers to form around the viral ori, which unwinds the dsDNA and starts the bi-directional duplication of the genome [36,39,52,53,56,60].

The LT is capable of multiple functions and is responsible for controlling much of the viral life and infectious cycle of MCPyV, ensuring viral survival [1]. Human cells have sensors like the Skp-F-box-cullin (SCF) E3 ubiquitin ligases (Fbw7, βTrCP, and Skp2) that can recognize conserved viral phosphorylation sites and mediate the degradation and repression of viral components. MCPyV LT was shown to have highly conserved phosphoprotein motifs that, upon serine/threonine kinase-mediated phosphorylation, provide a target for these E3 ligases [1,60]. This prevents its accumulation and replication in the cell, but at the same time, it may establish viral latency [1]. LT is continuously transcribed and translated but also immediately degraded so as not to reach the threshold required for the assembly of the multimeric LT helicase complex needed to replicate the entire viral genome [60]. However, under conditions of increased intracellular stress, such as nutrient loss, MCPyV responds with decreased activity of SCF E3 ligases, allowing LT to accumulate in the host cell [1]. By substituting alanine for the serine (S) residues S220 (Skp2 binding site) and S239 (Fbw7 binding site) in the LT phosphorylation sites, or by knocking down these SCF E3 ligases, Kwun et al. experimentally demonstrated both increased LT stability and LT-dependent viral replication [60]. Once a certain threshold is reached, the replication complex is assembled at the viral ori, with initial viral DNA synthesis and the subsequent expression of capsid proteins to form viral particles that can be released from the cell (Figure 2) [1,60,61]. The exact mechanism of MCPyV virion release from an infected cell is still not fully understood [61].

## 3. MCPyV-Mediated Carcinogenesis

Of all cancers worldwide, approximately 10–15% are caused by viruses [62]. DNA viruses known to cause cancer include the human papillomavirus (HPV), hepatitis B virus (HBV), Kaposi’s sarcoma herpesvirus (KSHV), Epstein–Barr virus (EBV), and MCPyV [62]. In the past, it has been demonstrated that these viruses encode oncoproteins with different mechanisms, commonly leading to the inactivation of the key tumor suppressor genes p53 and/or pRb. By disrupting the ability of p53 and pRb to provide genome stability, induce apoptosis, and prevent uncontrolled proliferation by controlling the cell cycle, the viral oncoproteins can ultimately lead to cancer [62]. Furthermore, deregulated oncogene expression was found to be associated with viral transformation [8,18]. While the MCPyV persists latently in an episomal form for years in healthy immunocompetent individuals without causing cytopathic effects to the host cell, data obtained from immunocompromised patients suggest the reactivation of the virus with subsequent mutagenesis and genomic integration [19]. When comparing serum anti-MCPyV capsid IgG antibodies, significantly increased levels were measured in MCC patients (88%) compared to those in healthy individuals (53%) [28,63]. This suggests that a resurgence of viral replication precedes the development of tumors [28]. However, no non-cancerous disease has been associated with an actively replicating virus, even in severely immunocompromised patients, such as individuals with AIDS. This rules out the presence of a high viral load as a causative agent of MCC [28,60].

Transcriptome analyses of samples from MCC patients have demonstrated the presence of monoclonally integrated viral dsDNA in the tumor cells’ genome [1,19,30]. In addition, a comparison of the viral genomes of MCC patients and MCC-derived cell lines and controls revealed tumor-specific mutations in the TA locus [19,64]. It is now generally assumed that MCC carcinogenesis is induced by the integration of the viral genome of MCPyV into the host DNA, as well as the expression of a truncated form of the viral oncogene LT [65,66]. The mutation limits uncontrolled viral replication, facilitates cell proliferation, and induces tumor development [1,32,66,67]. The low probability of these events happening both in one cell might explain the low incidence of MCC cases despite the ubiquitous prevalence of MCPyV in some populations [5].

### 3.1. Truncating Mutations in the Large T Antigen

The specific truncation of the LT in MCC cells is generated either by the integration of internal breakpoints or by nonsense mutations that generate stop codons, which both result in a loss of the original LT C-terminus. The majority of these mutations lie within the region between amino acids 252 and 431 [68]. In MCPyV-positive MCC, tumor growth is critically dependent on the presence of the TAs, especially LT [1,5]. The role of sT in cell proliferation is still under debate [69,70].

Although the regulation of TA expression has not been fully elucidated, reporter experiments showed variations in promoter activity depending on the presence of the truncating mutation in LT. While reduced promoter activity was observed with full-length LT, an increase in promoter activity was observed when the LT was truncated, as is the case in MCC [1]. A common consequence of these mutations is the absence of both the OBD and the ATP-dependent helicase domain (Figure 3) [66,67,69]. This allows for the stable integration of MCPyV DNA into the host genome and, most importantly, the expression of viral proteins without triggering cell cycle arrest by avoiding the activation of p53 and facilitating cell proliferation by binding pRb [67,71].

In approximately 75% of cancer types and about 50% of all cancers analyzed, mutations associated with the apolipoprotein B mRNA-editing enzyme catalytic polypeptide (APOBEC; A) represent one of the most common endogenous mutational mechanisms in cancer [72]. Soikkeli et al. investigated the contribution of activation-induced cytidine deaminase (AID) and APOBEC as a cause for the truncation of large T antigen in virus-positive MCC. By analyzing the mutations in MCPyV LT sequences of healthy and MCC samples, they showed significantly enriched APOBEC3 mutation signatures in MCC sequences [67].

The human APOBEC family of cytidine deaminases comprises eleven enzymes, namely, A1, A2, A3A, A3B, A3C, A3D, A3F, A3G, A3H, A4, and AID [73,74]. While AID is well known for its high expression in germinal center (GC) B cells due to its involvement in secondary antibody diversification, the APOBEC3 (A3) subfamily in particular has gained prominence in antiviral immunity [72,73]. With the exception of A2 and A4, a common function of the members of the APOBEC family is their ability to deaminate cytidine bases to uracil bases (C-to-U transitions). AID targets single-stranded DNA (ssDNA) and the A3 subfamily edits RNA or ssDNA, preferably mutating cytidines in a TC and CC dinucleotide constellation [72,73,75]. In a hydrolytic mechanism, the amine group (NH_2_) of cytidine is replaced by a carbonyl group (double-bonded oxygen), preventing the base from forming a trivalent bond with guanine but allowing it to bind adenine [76]. The deamination results in U:G mispairing, which can be highly mutagenic if they remain in the DNA and are not successfully excised by DNA repair mechanisms such as base excision repair (BER), resulting in C:G-to-T:A conversions [77]. However, the excision of uracil bases by uracil DNA glycosylase (UNG) potentially leads to abasic sites which, if not properly repaired, can lead to a further mutagenic event [77]. This harbors the risk of missense mutations and random stop codons truncating the sequence.

Although AID could be linked with carcinogenesis in several cancers, including different skin cancers such as squamous cell carcinoma, basal cell carcinoma, and melanoma, no significant role in MCC tumorigenesis has been attributed to AID [67]. However, the possibility that APOBEC3 plays an important role in MCC tumorigenesis has already been suggested after Verhalen et al. showed that A3B expression and activity are upregulated upon various polyomavirus infections, especially with BKPyV [78]. There, the LT alone was described to be sufficient to induce this response, implying that A3B expression is a consequence of viral infection [67,78,79]. When comparing the percentage of RNA expression levels of APOBEC3 subtypes (A3A, A3B, A3C, A3D, A3F, A3G, A3H) between MCPyV^+^ and MCPyV^−^ groups, a higher expression was registered for A3A (1.2-fold change), A3D (3.4-fold change), and A3H (4.0-fold change) [67]. However, A3B, A3C, A3F, and A3G RNA expression levels were either similar or higher in MCPyV-negative samples [67].

The comparison of SV40 with other known polyomaviruses has consistently shown that BKPyV and SV40 are closely related and highly homologous in their molecular mechanisms and interactions, whereas both are phylogenetically distant from MCPyV, which is particularly noticeable when considering interactions with key oncogenic proteins such as p53 and pRb [5,80]. This shows that it is difficult to generalize certain findings between polyomaviruses. As MCC is a very rare tumor, much more investigation is needed to find targets that may have a reciprocal relationship with APOBEC3 and in combination lead to tumorigenesis. Also, despite the observation that strong APOBEC3 mutation signatures have been observed in the LT region of MCPyV-positive MCCs, the role of APOBEC3 in carcinogenesis is still debated, and further research is needed, as no other studies have yet validated the same pattern.

While most evidence argues that mutation of the viral genome occurs prior to or during integration into the human genome [19], there is also the possibility that a truncated version of LT results from alternative splicing of the full length LT mRNA. Although it has been shown that an alternatively spliced form of the BKPyV LT has transforming properties [81], the isoforms of the TA of MCPyV (Figure 1), namely, sT and 57 kT, are not oncogenic [36,37,64,82,83]. Therefore, while it is theoretically possible that alternative splicing could contribute to LT truncation, there is no evidence to confirm that alternative splicing of the viral TA leads to tumorigenesis.

### 3.2. Somatic Integration of MCPyV Viral DNA

As a second requirement for MCPyV-induced tumorigenesis, the mutated viral genome has to be integrated into the host genome [8]. The presence of monoclonally integrated MCPyV in MCC cell genomes gave reason to assume that the chromosomal integration precedes the clonal expansion of the tumor cells, constituting a founder event in oncogenic transformation [84]. Unlike other DNA viruses, MCPyV integrates in a manner that selectively preserves the function of specific genetic elements—notably, the early gene promoter, allowing for the expression of sT and truncLT [30].

Although it is suggested that viral genome integration and LT truncation occur independently, the question of whether LT integration or mutation occurs first in MCC is still debatable [19,85]. There are cases indicating that LT truncation might happen while being integrated into the host genome [85]. By working with MCPyV-positive MCC cell lines, it could be demonstrated that an incomplete integration of MCPyV DNA led to the truncation of the LT in one out of six cell lines [85]. Using multiple sequencing approaches, Czech-Sioli et al. further provided evidence that LT mutagenesis occurs prior to or during the integration of the viral genome, not afterwards [19]. They attribute the expression of MCC-characteristic truncLT to point mutations in the DNA sequence or the occurrence of deletions within LT. These cause premature stop codons and frameshifts, respectively, leading to truncLT in all cell lines, with the LxCxE motif being preserved, as is known from clinical samples [19].

Furthermore, it has been shown that integration occurs in predominantly open chromatin, with more integration sites found in introns but also intergenic or centromeric regions and chromosome 5 being prone for integration events [19,86,87]. Also, the integration events can be divided into two groups based on how their genomic rearrangement presents [19]. The first group involves the viral genome integrating linearly by non-homologous end joining (NHEJ) as a single or concatemeric sequence with host junctions flanking the integrated fragment [19]. These junctions characteristically map in the vicinity of one another, which indicates that the integration of the virus did not cause major deletions or amplifications of the host sequences [19,85,86]. The second group is distinguished by the presence of amplifications of the flanking cellular DNA, which result from microhomology-mediated break-induced replication (MMBIR), causing the virus–host junctions to be separated by thousands of base pairs [19,86]. This occurrence argues for large rearrangements that happen in the human genome [19,86,88]. In line with this, Schrama et al. were able to show sequence rearrangement in a cell line, which caused the generation of truncLT during the process [85]. However, Harms et al. suggest that integration in the host genome happens due to the random fragmentation of the genome with no specific integration sites [5].

### 3.3. Carcinogenic Effects of truncLT

The continuous expression of the truncLT is required for MCC cell lines to maintain their proliferative phenotype [1,8,11]. Various oncogenic functions have been attributed to the protein, which are described below. The ability to bypass cell-cycle control through the interaction of oncogenes with tumor suppressors is known from various oncogenic viruses. The two major tumor suppressors are pRb and p53, which are known to lose their control over cell proliferation upon malfunctional activity [36]. These have also been attributed to the MCPyV LT.

#### 3.3.1. Disruption of the Binding of pRb and E2F Transcription Factors

Located N-terminally of the truncating mutations, the LxCxE motif remains unaffected in truncLT [1]. This allows truncLT to hijack the cell cycle machinery by binding and inactivating pRb [66,69], which is the major regulator of the G1/S transition in the cell cycle [49,89,90,91]. Until being phosphorylated by active cyclin D-CDK4/6, pRb binds and thus inactivates the cell cycle-activating E2F transcription factors. This interaction leads to transcriptional repression mechanisms such as the recruitment of histone deacetylases (HDACs), causing E2F-dependent promotors to be restrained (Figure 4A) [49]. This prevents the expression of E2F target genes, which are then unable to initiate DNA replication [89]. As a consequence, the interactions between pRb and E2F prevent entry into the cell cycle [90]. In contrast, growth factors mediate the phosphorylation of pRb by CDK4/6, which causes its dissociation from E2Fs (Figure 4A) [36,89]. This allows for the transcription of genes driving cell cycle progression and cell proliferation, highlighting the phosphorylation state of pRb as a major event within the cell cycle [36,89].

Consequently, mechanisms leading to the constant inactivation of pRb may cause the permanent loss of its tumor-suppressing function, leading to uncontrolled cell proliferation and tumor formation. Overall, the loss of pRb function strongly coincides with poor progression-free survival (PFS), disease-specific survival (DSS), and overall survival (OS) in several cancer types [92].

Upon infection with MCPyV, the LT plays an important role in short-circuiting the repression of the E2F pathway by binding with high affinity to pRb via the conserved LxCxE motif, preventing it from regulating E2F [90]. The C-terminal OBD and helicase domain of full-length LT allow for viral replication and persistent viral TA expression. Li et al. showed that high levels of MCPyV LT activate the DNA damage response, leading to increased p53 phosphorylation and the activation of p53 downstream target genes [93]. This results in the induction of cell cycle arrest and the inhibition of cell proliferation but also in the production of new virions (Figure 4B). Contrary to an accumulation of LT in the cell, low levels of LT avoid the activation of p53 while still promoting cell cycle progression. This allows the virus to induce latency in the cell [60]. The regulation of the levels of LT allows the virus to react to the viability of the host cell and ensures its survival and persistent infection [1,60]. In MCC, however, the truncation of LT results in a loss of the C terminal domain, rendering the virus incapable of transmissible replication. Furthermore, the LxCxE motif remains intact, allowing truncLT to bind pRb with high affinity (Figure 4C) [31]. The deletion of the C-terminus prevents p53 activation, which would lead to cell cycle arrest and therefore further support tumor growth [31]. Consequently, the absence of the activation of DNA damage response pathways allows pRb to be permanently inactivated by truncLT, allowing E2F to enter the S phase without restraint, resulting in the expression of uncontrolled cell cycle genes leading to cell cycle progression and proliferation [31].

Both non-tumorous and tumor-derived LT interact with pRb, while the mutagenesis of the retinoblastoma protein binding motif (LxCxE to LxCxK) in vitro results in failed pRb-LT complex formation [32,64]. The interaction disrupted the regulation of E2F by pRb, causing the transcription factors of the E2F family to be transcriptionally more active and ultimately leading to increased proliferation by expressing genes required for cell cycle progression [32]. The importance of the interaction between truncLT and pRb has also been demonstrated in mouse-models, where the specific knockdown of LT, and thus no inactivation of pRb, resulted in an inhibited proliferation of MCPyV^+^ MCC [50]. A re-expression of LT rescued the phenotype in a cell line model [50]. Therefore, the binding of pRb and LT is critically required for tumor growth [50].

#### 3.3.2. Interaction with p53

It has been consistently shown in MCPyV^+^ individuals that full-length LT with functional helicase activity induces cell cycle arrest, leading to limited cell proliferation in a p53-dependent manner [30,93]. Although the DNA damage response-activating domain is deleted in truncLT, it was shown to indirectly activate p53 [94]. This is in contrast to other polyomaviruses that inactivate p53 by directly binding to the LT in addition to pRb to promote cellular transformation [30,59,95]. While many cancers can be directly linked to inactivating p53 mutations, in MCC, truncLT-pRb binding leads to an increased expression of ARF (alternate reading frame tumor-suppressor protein) [94]. This tumor suppressor is an inhibitor of MDM2 (mouse double minute 2), a p53 E3 ubiquitin ligase that suppresses p53 activity [31,36]. The lack of MDM2 activity leads to the maintenance and indirect activation of p53 [30,32,96]. Nevertheless, Houben et al. showed that virus-positive MCC lines express p53 with only limited activity [97]. Therefore, the tumor promoting the lack of p53 activity can be attributed to LT-independent pathways.

#### 3.3.3. Deregulation of mi-375 by Protein Atonal Homolog 1 (ATOH1)

Malignant transformation in different cancers is commonly accompanied by an altered expression of microRNAs (miRNAs) [98,99,100]. miRNAs comprise small (21 to 25 nucleotides) non-coding RNAs exerting the posttranscriptional repression of gene expression and are therefore crucial for biological processes [100,101]. Comparing miRNA expression profiles of both virus-negative and virus-positive MCC, MicroRNA375 (miR-375) was found to be deregulated, with overexpression in the latter but not in virus-negative MCC, normal skin, and other non-MCC skin cancers [98,99,101]. In fact, miR-375 was identified as the most highly expressed miRNA in MCC cell lines and tissues [101,102,103]. Functionally, miR-375 is a tumor suppressor inhibiting cell proliferation, migration, invasion, and tumor metastasis by targeting oncogenes [98]. Kumar et al. showed that lactate dehydrogenase B (LDHB) is an important target of miR-375 in virus-positive MCC [98]. LDHB is a glycolytic enzyme that catalyzes the conversion of lactate to pyruvate, but also NAD^+^ to NADH, which are substrates for the Krebs cycle [98]. This mechanism was repressed by miR-375 in MCPyV^+^ MCC, suggesting that aerobic glycolysis is required for MCPyV^+^ MCC growth, which depends on the generation of NAD^+^, which is maintained by LDHB repression [98]. Conversely, miR-375 suppression inhibited cell growth in MCPyV^+^ MCC cells, leading to the conclusion that miR-375 is critical for cell viability in virus-positive cells and acts as an oncogene in virus-positive MCC [98].

Fan et al. showed a positive correlation between protein atonal homolog 1 (ATOH1) and miR-375 expression levels. ATOH1 is a master regulator of Merkel cell development [104], whose importance in MCC tumorigenesis was shown by co-expressing MCPyV Tantigens with ATOH1 in mice, which resulted in MCC-like tumors [105]. Additionally, truncLT was found to stimulate the expression of ATOH1 [99]. ATOH1 expression was negatively correlated with DNA methylation in the ATOH1 promoter region, with hypomethylated CpG islands in MCC cell lines and tissues [99]. ATOH1 belongs to the basic helix–loop–helix family of transcription factors, activating transcription by binding to E-boxes (5′-CANNTG-3′), which can be commonly found in the miR-375 promoter [99]. Moreover, reporter assays showed that ATOH1 was capable of activating the miR-375 promoter [99]. Indeed, truncLT was able to trigger the ATOH1/miR-375 expression cascade while sT did not [99]. Subsequently, Harold et al. attributed ATOH1 expression to TA-induced Sox2 expression [106]. This induction was critically dependent on the Rb-binding site of TA, while the demethylation of both the ATOH1 and Sox2 promotors was only observed in MCPyV-positive MCC cell lines [106]. Still, it should be noted that this relationship is still controversial. For example, Fu et al. report a positive correlation between downregulated protein expression levels of ATOH1 and MCC recurrence [107]. Therefore, further studies in larger cohorts are needed to clarify this relationship.

## 4. Immune Escape Mechanisms of MCPyV

The dependency of the expression of the viral oncogenes by MCC leads to the assumption that the immune system is able to detect the viral antigen epitopes on the malignant cells [108]. The fact that it is not only immunocompromised people who are developing this aggressive skin tumor but also that the majority of all MCC patients possess a functional immune system indicates that the tumor cells must have the capacity to evade the immune system efficiently [108,109,110]. MCC uses various principles of immune escape, as reviewed by Jani et al. [108]. Here, we shortly want to highlight two mechanisms, the downregulation of toll-like receptor (TLR) 9 and MHC class I molecules on the surface of the tumor cells.

### 4.1. Downregulation of TLR9 Maintains Chronic Infection of the Cell

As part of the innate immune system, cells are equipped with various pattern recognition receptors (PRRs), such as TLRs, which recognize pathogen-associated molecular patterns (PAMPs) to trigger an initial immune response even before a stable infection is established [111,112,113].

Mainly expressed in plasmacytoid dendritic cells (pDCs), TLR9 recognizes phagocytosed foreign intracellular dsDNA with unmethylated 2′-desoxyribo (cytidine—phosphate—guanosine) (CpG) motifs such as those found in the genome of viruses like MCPyV [113,114,115,116]. After recognition, large quantities of type I interferons are secreted, and the expression of the transcription factor NF-κB is induced, which leads to the production of inflammatory mediators like interleukin (IL)-6, IL-8, and tumor necrosis factor α (TNFα) [113,114,115]. The expression of TLR9 is positively regulated by the transactivator CCAAT/enhancer-binding protein (C/EBP) β, which binds the C/EBPβ response element in the TLR9 promoter [108,113,117,118]. Notably, C/EBPβ is a key player in interferon signaling by forming complexes with other transcription factors such as NF-κB, but also SP-1 and STAT3, needed to induce the transcription of IL-6, IL-8, and TNFα, all necessary to mediate acute inflammation against invaders [119].

Shahzad et al. were able to show that MCPyV LT severely disrupts C/EBPβ binding and activity by downregulating its mRNA levels [113]. The subsequent downregulation of TLR9 and other immune-related proteins like IL-13, IL-24, and CD114 may reduce the recruitment of immune cells, providing innate protection upon MCPyV infection [108,113]. Due to the similarity to other oncogenic dsDNA viruses like EBV, HBV, and HPV16, the downregulation of TLR9 by MCPyV has been suggested as an immune escape mechanism, which also supports the malignant MCPyV-infected cells to avoid the effective infiltration of anti-tumor immune cells.

### 4.2. MHC-Dependent Immune Escape

MCC tumors and MCC cell lines are known to downregulate MHC class I expression, another reason for inadequate T-cell infiltration, and this is evidence that MCC prevents the involvement of the adaptive immune system [120]. In fact, this downregulation has been detected in 84% of MCCs [121,122]. This impairs the recognition of tumor cells by cytotoxic T lymphocytes, allowing the cancer cells to escape immune surveillance [122,123].

Both LT and sT contain several T-cell epitopes that can be recognized by T cells from both patients and healthy donors [120,124,125,126,127]. While responses to the viral protein VP1 were found in both healthy donors and MCC-patients, responses to the LT and sT were clearly increased in patients [120,127]. T-cell responses against MCC LT have been thoroughly reviewed elsewhere [128].

Lee et al. assumed that MCPyV oncoproteins are responsible for the MHC I downregulation [122]. Inactivating both viral oncogenes (sT and LT) using shRNA resulted in the upregulation of class I genes, including >1.5-fold increases in HLA-B and HLA-C [122]. Furthermore, they showed the transcription factor MYC homolog MYCL (L-MYC) to be the mediator of MHC-I suppression [122]. Notably, sT but not LT was shown to bind and recruit MYCL to the EP400 histone acetyltransferase and chromatin modifier complex [129]. A knockdown of EP400 resulted in a threefold increased level of HLA-B and HLA-C [122]. This indicates that the MCPyV oncogene sT is involved in MHC-I downregulation, while the involvement of LT in this process remains enigmatic. Sauerer et al. showed that IFNγ-treatment was able to restore the expression of HLA-A, -B, and -C in MCC cell lines while not changing the expression level of the truncLT [130].

In addition to MHC downregulation, the persistent presentation of LT antigen on MHC-I molecules may lead to dysfunctional T cells [131]. It has been shown that MCC tissue harbors particularly LT-specific CD8^+^ T cells, which are, especially inside the tumor, often exhausted [126,132]. This may be another strategy employed by the virus to evade reactivity to LT, leading to chronic infection and, ultimately, immune tolerance [126,132,133].

## 5. LT-Based Treatment Strategies in MCC

Despite its rare occurrence and it initially being often misdiagnosed as a benign nodule, MCC is a highly aggressive cutaneous malignancy with strong recurrence and metastatic potential, causing the course of the disease to be very challenging and exhausting [134,135,136]. The 5-year survival rates for MCC patients are 51%, 35%, and 14% for localized, nodal, and metastatic stages, respectively, and the case-by-case fatality rate is the highest among all skin cancers [1,2,3,4,5,134,137]. Although these data come from the pre-immune checkpoint era, the rising incidence of MCC counterweights the clinical successes. With 2488 cases in the USA in 2013 and an increase in the incidence rate, it is estimated that there will be 3200 cases of MCC in 2025 [108,135]. The reasons for this increase are not yet clear, but an aging population and improvements in diagnostics have been suggested [5,135].

Although great progress has been made in understanding the molecular mechanisms of LT in MCC, it remains a highly aggressive disease with limited curative options. Immune checkpoints like the PD-1 (programmed death 1) and its ligand, as well as CTLA4 (cytotoxic T lymphocyte-associated antigen 4), are targets of immunotherapy and have become part of the standard treatment for MCC [108,128,130,136,138,139,140,141,142]. Further clinical studies investigate the potential beneficial efficacy of combining immune checkpoint inhibitors [143,144]. Despite the recent success, alternative treatment strategies are needed [134,145], and targeting the MCPyV LT appears suitable for tackling the MCC-specific tumorigenic principle. One possibility would be targeting the interaction of the LT and RB with a small molecule inhibitor. Such an inhibitor has been described for the RB-binding protein E7 from human papilloma virus as an antiviral reagent [146]. However, until now, to our knowledge, no such approach has been reported for MCPyV. In MCC therapy, all hitherto reported LT-specific treatment strategies focused on exploiting immune responses against the viral oncogene, with a focus on T-cell responses [128].

### 5.1. T-Cell-Based Immunotherapy

Regardless of the stage of diagnosis, functional tumor-infiltrating CD8^+^ T cells (TILs) have been associated with improved prognosis and survival in MCC patients, especially in MCPyV-positive patients [30,108,128,136,137,147,148,149]. These TILs, which have already encountered the tumor-specific LT antigen in the patient, can be isolated and expanded ex vivo with IL-2 before being reinfused into the same patient [150]. TIL-based therapy takes advantage of the natural ability of T cells to recognize and attack cancer cells with increased numbers and potency, bypassing the state of T-cell exhaustion and anergy caused by the immunosuppressive tumor microenvironment (TME) [132,137,149,151,152]. Nevertheless, Iyer et al. point out that the constant presence of the antigen in the MCC tumor leads to an exhausted T-cell phenotype due to high levels of the immunosuppressive cytokine IL-10 and the presence of FoxP3^+^ T_regs_, which promoted tumor growth [124]. Consequently, Ryu et al. found PD-1 and CD39 to be expressed on MCPyV-specific CD8^+^ T cells, with CD39 being commonly found on exhausted CD8^+^ TILs [153]. Samimi et al. further point out that the T-cell-mediated pressure on the MCC tumor can be the reason for the transcriptional loss of HLA molecules on the cancer cells [131]. These findings indicate that a T-cell immunotherapy based on TILs could have potential efficacy, but the only clinical study that treats advanced MCC patients with TILs is still recruiting, and primary results are not expected until 2036 (NCT03935893).

Although TIL-based T-cell therapy seems promising, CD8^+^ T-cell activation is strongly supported by CD4^+^ T helper (Th) cells in the TME [137,154,155,156]. Therefore, a combination of both T-cell subsets in T-cell therapy similar to treatments for malignant melanoma has been proposed for MCC [154,157,158]. Longino et al. identified LT-specific epitopes for CD4^+^ T cells that are assumed to occur naturally in virus-positive MCC in vivo [137,154]. One of these epitopes, WED_LT209–228_, includes the LxCxE motif, which is critical for the oncogenic activity of LT [64,132,154,159]. In summary, data generated in the previous years indicate the therapeutic value of TILs in MCC but at the same time highlight the need to overcome the suppressive environment within the tumor and the exhaustion of the tumor-specific T cells.

### 5.2. Large T Antigen-Based Vaccine Approaches

Besides the adoptive transfer of T cells, a therapeutic vaccination against the LT of the MCPyV could induce an effective immune response against the tumor driver of MCC, inducing not only cellular but also humoral responses [160]. Since antibodies against LT could only be detected in MCPyV^+^ MCC tumors and not in MCPyV^+^ individuals, and since the absence of MCPyV antibodies in MCC patients correlates with a worse prognosis, it is suggested that LT could be a possible antibody target [161,162]. However, the effects of a humoral response against an intracellular antigen are unclear.

More research has been conducted on the induction of cellular immune responses against LT and, to some extent, also sT, and about 30 epitopes have been reported so far [161,162]. Although the primary target of therapeutic vaccines is the induction of tumor-specific cytotoxic CD8^+^ T cells, the role of CD4^+^ T cells has become increasingly considered. Since the latter are required to induce memory-type CD8^+^ T cells [163], the identification of MHC class II epitopes and ways to facilitate MHC class II-restricted presentation have been focused on and are described below. The topic of vaccines against MCC has also been recently reviewed in more detail by Gambichler et al. [164].

Zeng et al. tested a DNA vaccine against the N-terminal 258 amino acids of LT in a murine MCC tumor model and found antitumor effects that were mainly mediated by CD4^+^ T cells [125]. Gomez et al., from the same lab, showed strong CD8^+^ induction in the same model against LT-calreticulin fusion [165] and against sT [166].

However, when applying functional domains from oncogenes in the form of DNA, special care must be taken to prevent putative carcinogenic activity of the vaccine. In doing so, it is important to ensure that epitopes containing this motif do not lose their immunogenicity when mutated, especially if this motif is the main part of the epitope that has been shown to be crucial for the immune response. Schrama et al. present such a possibility for the LT by mutating the serine in position 220 to alanine (S220A), thus preventing serine phosphorylation, which is critical for the tumor-promoting function of the LT [159]. Buchta Rosean et al. designed a DNA cancer vaccine to enhance CD4^+^ T-cell responses [160]. The fusion of truncated LT^S220A^ with lysosomal-associated membrane protein 1 (LAMP1) was shown to specifically target the lysosome and enhance antigen presentation by MHC-II to mediate LT-specific Th1 cell responses (LT_145–157_). CD8^+^ T cells were also observed to be increasingly activated through vaccine-dependent enhanced MHC I presentation [154,159,160].

Although being of viral origin, research on LT-based vaccine approaches has shown that the truncated LT is only moderately immunogenic, which is why Gerer et al. used tumor antigen-loaded dendritic cells (DCs) to induce tumor-specific T cells [127,128]. DCs are the main inducers of adaptive immune responses and efficiently present antigenic peptides on MHC-I and MHC-II, thereby activating CD8^+^ and CD4^+^ T cells, which are then able to mount their effector functions [127,167,168]. Ex vivo-generated DCs, which are loaded with antigen by mRNA-transfection, have been used in various clinical trials [169]. When using DCs transfected with an LT-DCLamp fusion construct, which, like LAMP1, facilitates MHC class II-restricted presentation [170], the stimulation of mixed CD4^+^ and CD8^+^ T cells resulted in higher IFNγ secretion than with CD8^+^ T cells alone, again highlighting the relevance CD4^+^ helper T cells [127]. To prevent the development of escape mutants, it would be advantageous to target epitopes within functional domains of the LT. Longino et al. identified LT-specific epitopes on CD4^+^ T cells that are assumed to occur naturally in virus-positive MCC in vivo [154]. One of these epitopes, WED_LT209–228_, includes the LxCxE motif, which is critical for the oncogenic activity of LT [64,132,154,159].

Despite these encouraging preclinical reports, few clinical trials are performed on therapeutic vaccination against MCC. Of the nine ongoing clinical trials summarized by Gambichler et al., none use the LT as a specific antigen [164]. Only one trial (NCT05422781) is mentioned, which used the LAMP1-fusion of the tuncLT^S220A^ described by Buchta Rosean. This phase I trial with six patients was completed in June 2023, but no results have been revealed yet [160].

Overall, studies on TA-based vaccination approaches show a high potential to induce directed immune responses in preclinical experiments, and the transition to clinical trials has started.

## 6. Conclusions

As a key player in the development and progression of MCC, the LT antigen of the MCPyV is the major driver of this rare and highly aggressive skin cancer. For an oncogenic transformation, the viral genome has to integrate into the human genome, and a specific truncation mutation has to be expressed. The truncLT inactivates pRb, leading to uncontrolled cell proliferation, but several accompanying effects appear relevant for tumor progression as well, including immune escape mechanisms and miRNA deregulation. Al-though treatment strategies based on the tumor-specific LT seem promising, further research has to be conducted to map the downstream signaling network and the antigenic potential of this viral oncogene for a new effective treatment option for MCC patients.

## Figures and Tables

**Figure 1 genes-15-01127-f001:**
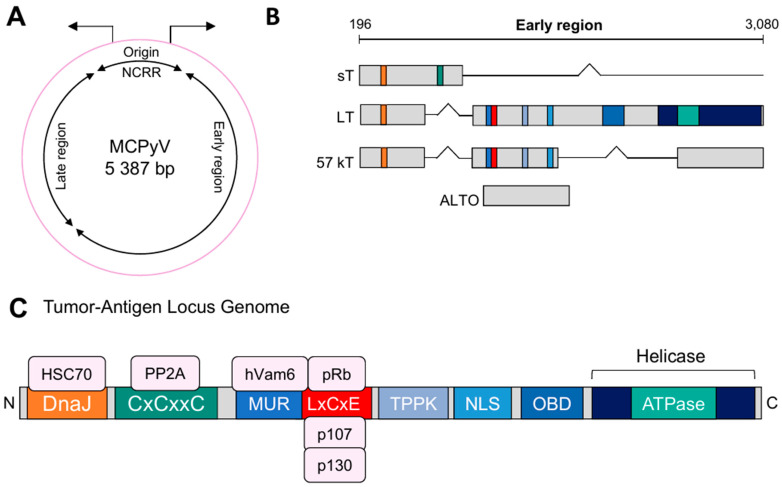
Organization of the MCPyV genome. (**A**) The map of the circular genome shows the genes of the early and late region. (**B**) The tumor antigen (TA) gene can be alternatively spliced into the small TA (sT), the large TA (LT), a 57 kT transcript, and an alternate reading frame called ALTO. (**C**) The TA locus consists of multiple functional domains that are able to interact with several proteins. The DnaJ domain at the N-terminus is bound by heat shock cognate 71 kDa protein (HSP70). Adjacent to the DnaJ domain are two cysteine (CxCxxC) motifs, here schematically shown as one, which are only expressed in the sT transcript and bind protein phosphatase 2A (PP2A). The domain next to the CxCxxC motifs is the MCPyV unique region (MUR) that cooperates with the endosomal protein hVam6. The retinoblastoma (Rb) binding motif (LxCxE) is required for LT to bind the Rb protein (pRb). The LxCxE motif is also bound by the pRb-related proteins p130 (RBL2) and p107 (RBL1). The domain adjacent to the LxCxE motif contains a threonine-proline-proline-lysine (TPPK) motif, the nuclear localization signal (NLS), and the origin binding domain (OBD). The C-terminal helicase contains an ATP-binding domain.

**Figure 2 genes-15-01127-f002:**
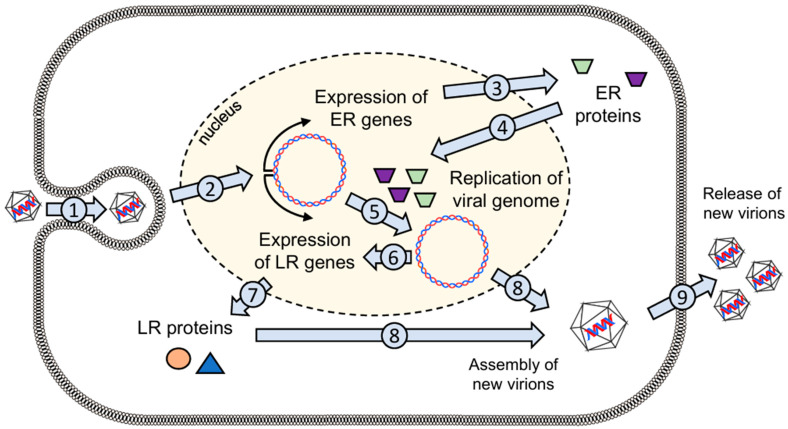
Viral life cycle of MCPyV. (1) The virus enters the cell via endocytosis and (2) the viral genome is transported to the nucleus. (3) The genes of the early region (ER) are transcribed and then translated into ER proteins in the cytoplasm. (4 and 5) The ER proteins induce the replication of the viral genome in the nucleus. (7) Afterwards, proteins of the late region (LR) are expressed. (8) Both LR proteins and newly synthesized viral DNA are used to assemble new virions that (9) are released from the cell.

**Figure 3 genes-15-01127-f003:**
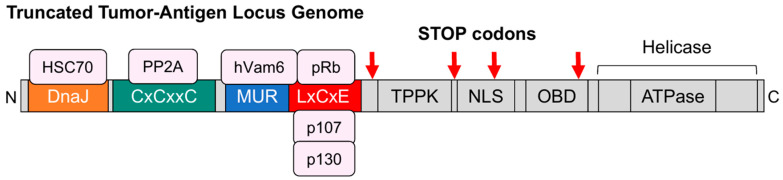
The gene locus of the oncogenic truncated large tumor-antigen (truncLT). The truncLT still contains the same functional domains in the N-terminus as the full-length large tumor-antigen including the DnaJ domain, cysteine motifs (CxCxxC), MCPyV unique region (MUR), LxCxE motif, binding heat shock cognate 71 kDa protein (HSC70), protein phosphatase 2A (PP2A), endosomal protein hVamp6, retinoblastoma protein (pRb), and pRb-related proteins p130 (RBL2) and p107 (RBL1), respectively. After the LxCxE motif, the truncLT has acquired stop codons that render the following nuclear localization signal (NLS), the origin binding domain (OBD), and the helicase futile.

**Figure 4 genes-15-01127-f004:**
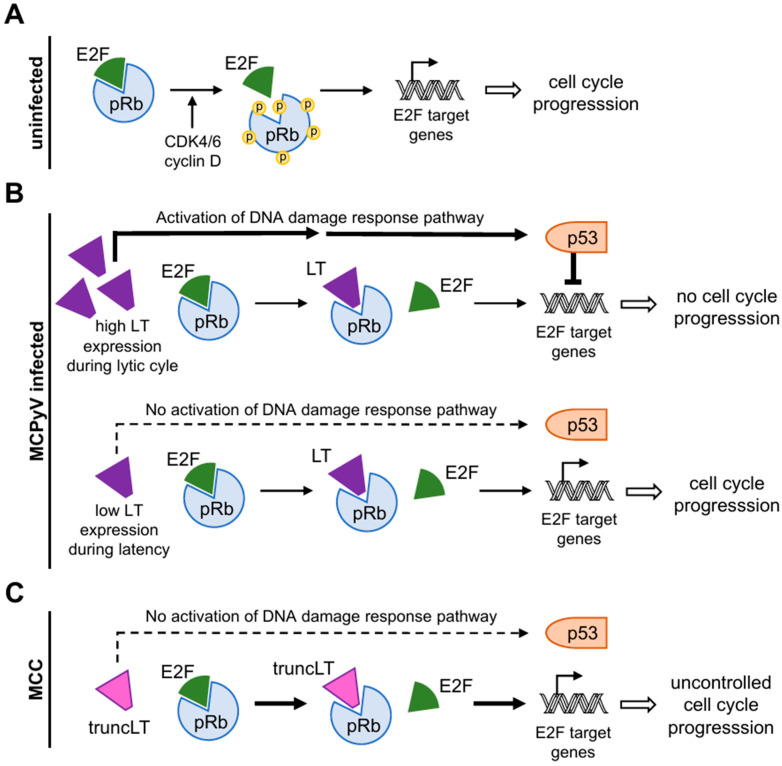
The influence of the large tumor-antigen (LT) and the truncated LT (truncLT) on the cell cycle progression via their interference with the retinoblastoma protein (pRb) and the E2F transcription factors. (**A**) In uninfected individuals, mitogenic signals activate the cyclin D/CDK4,6 complex which phosphorylates pRb. As a result, pRb dissociates from E2F, which induces cell cycle progression by the expression of E2F target genes. (**B**) In MCPyV-infected individuals, LT competes with E2F for the binding to pRb and can therefore induce the expression of E2F target genes. Nevertheless, the latent state and the lytic phase of the virus have different effects on cell proliferation. High levels of LT in the lytic state activate the DNA damage response pathway and, subsequently, the tumor suppressor p53, which inhibits cell proliferation. During latency, the LT levels are too low to activate p53 and the cell cycle can progress. (**C**) In MCC patients, the truncated LT (truncLT) is unable to trigger the DNA damage pathway and simultaneously constitutively inhibits the binding of pRb and E2F. This leads to the expression of E2F target genes, which results in uncontrolled cell proliferation.

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
