# Peer review of "The Role of the Large T Antigen in the Molecular Pathogenesis of Merkel Cell Carcinoma"

_genes, 2024, doi:10.3390/genes15091127_

Round 1
Reviewer 1 Report
Comments and Suggestions for Authors
Dear Authors,
I was pleased to read this interesting review.
The manuscript is the newest in his field, is clearly structured and cites the most recent literature relevant to the topic and also MCC is an excellent model for answering open questions in cancer immunology and immunotherapy.
As regarding the drafting the text:
Ø the insertion of the figures in the text is done without bold
Ø the abbreviation of the journaÈ™ is made with a period between words

Author Response
First of all, we thank the reviewer for reading and evaluating our manuscript.
The manuscript is the newest in his field, is clearly structured and cites the most recent literature relevant to the topic and also MCC is an excellent model for answering open questions in cancer immunology and immunotherapy.
Thank you for this kind evaluation.
As regarding the drafting the text:
Ø the insertion of the figures in the text is done without bold
Answer: Thank you for mentioning this. We have revised the manuscript and removed the bold formation of the references to the figures.
Ø the abbreviation of the journalÈ™ is made with a period between words
Answer: We have adjusted the references and the abbreviations of the journals include now periods between the words.

Reviewer 2 Report
Comments and Suggestions for Authors
In this review, Myrda and colleagues compile an excellent, comprehensive, and exhaustive revision of the scientific literature on the role of Large T-Antigen (LT) in the pathogenesis of Merkel Cell Carcinoma (MCC). From this review you can learn that the clonal integration of Merker Cells Polyomavirus (MCPyV) DNA into the host genome is a crucial event in promoting this malignancy. Moreover, the expression of truncated forms of the LT (truncLT), seems directly contribute to carcinogenesis. The truncation of the LT’s C-terminus prevents the virus from replicating, precluding cytopathic effects that would lead to cell death. truncLT exerts is oncogenic role by binding and inactivating the retinoblastoma protein (pRb), which results in cell proliferation and tumor development. Given its leading role in MCC, the LT can be a valid elective target for therapeutic interventions with novel approaches, such as T cell-based immunotherapy and cancer vaccines. The review is well structured, and it can be accepted for publication after a minor editing of English language and correction of same typo mistakes.

Minor editing of English language and correction of same typo mistakes
Author Response
First of all, we thank the reviewer for reading and evaluating our manuscript.
In this review, Myrda and colleagues compile an excellent, comprehensive, and exhaustive revision of the scientific literature on the role of Large T-Antigen (LT) in the pathogenesis of Merkel Cell Carcinoma (MCC). From this review you can learn that the clonal integration of Merkel Cells Polyomavirus (MCPyV) DNA into the host genome is a crucial event in promoting this malignancy. Moreover, the expression of truncated forms of the LT (truncLT), seems directly contribute to carcinogenesis. The truncation of the LT’s C-terminus prevents the virus from replicating, precluding cytopathic effects that would lead to cell death. truncLT exerts is oncogenic role by binding and inactivating the retinoblastoma protein (pRb), which results in cell proliferation and tumor development. Given its leading role in MCC, the LT can be a valid elective target for therapeutic interventions with novel approaches, such as T cell-based immunotherapy and cancer vaccines. The review is well structured, and it can be accepted for publication after a minor editing of English language and correction of same typo mistakes.
Answer: We have revised the manuscript focusing on editing the English language as well as on typo mistakes and consistent abbreviations throughout the manuscript, using the track-changes-function in word.
